# A Pilot Study on a Nurse Rehabilitation Program: Could It Be Applied to COVID-19 Patients?

**DOI:** 10.3390/ijerph192114365

**Published:** 2022-11-02

**Authors:** José Moreira, Pedro Fonseca, Susana Miguel

**Affiliations:** 1Nursing Department, University of Évora, Largo do Senhor da Pobreza, 7000-811 Évora, Portugal; 2Comprehensive Health Research Centre, 1600-560 Lisbon, Portugal; 3National School of Public Health, 1600-560 Lisbon, Portugal; 4USF Alcaides, 7050-155 Montemor-o-Novo, Portugal; 5Department of Head and Neck Surgery and Endocrinology, Portuguese Institute of Oncology, 1099-023 Lisbon, Portugal

**Keywords:** coronavirus, rehabilitation nursing, respiratory tract diseases, breathing exercises, quality of life

## Abstract

(1) Background: An aging population, pollution and an increase in life habits that are harmful to respiratory health, and more recently the COVID-19 pandemic, have led to an increase in chronic respiratory diseases. Thus, this pilot study aims to describe an intervention program on the training of respiratory patients to maintain airway permeability and preventing complications. (2) Methods: An observational, analytical, and prospective study was proposed. After the initial evaluation of each user during hospitalization, the program started with two sessions, at discharge for the second phase, and fifteen days after discharge for the third phase. Throughout the program’s implementation, the modified Medical Research Council scale and the Barthel Index were applied. (3) Results: The population studied aged between 39 and 76 years, diagnosed with pneumonia or chronic obstructive pulmonary disease, showed a significant improvement in the degrees of dyspnea and levels of functionality, as well as an adequate level of learning. (4) Conclusions: This program positively impacted the patients’ independence by reducing dyspnea and increasing functionality of the patients included in this study. The set of exercises and techniques can be replicated at home and may be fundamental in the management of respective recovery, as well as in the COVID-19 pandemic.

## 1. Introduction

Currently, respiratory diseases assume a high economic weight in developed countries, given their incidence and prevalence, leading to high hospital admissions and deaths/year [1]. Pulmonary rehabilitation has been shown to improve patient awareness, improve respiratory function, and reduce the risk of complications, length of hospital stays, mortality rates, and re-admission risks [2]. Severe COVID-19 infection often leads to impairments requiring pulmonary rehabilitation at all phases of recovery [3].

To combat the spread of COVID-19, different scientific societies and disease control centers have promoted the uptake of preventive behaviors, such as using telemedicine and avoiding non-essential visits to health units. Adherence to these recommendations may pose a challenge among individuals with chronic respiratory conditions, given their increased healthcare needs compared to the general population [4]. Given this context, we can affirm that rehabilitative care in this type of gradually ageing population and patients with chronic respiratory conditions is a priority need, as well as an urgent need in society [5].

Rehabilitation, from the perspective of training, involves actively participating in all stages that people go through until they can fully meet their needs and/or find compensatory mechanisms to do so, with the rehabilitation nurse specialist being the guide in this process [5]. The rehabilitation nurse can and should play a significant role in the training of the respiratory patient through preparation programs for clinical discharge, endowing the patient with knowledge and techniques that allow them to manage their pathology [1].

Health units receive patients with a wide variety of diagnoses, namely: respiratory infections (community-acquired pneumonia (CAP)), pleural effusions, chronic obstructive pulmonary disease (COPD), severe dyspnea, pulmonary thromboembolism, and acute lung edema [1]. For most of these pathologies, intervention needs to arise as early as possible because, in most cases, to restore an adequate state of functionality, it is not enough to treat the pathology, but it is necessary to rehabilitate the patient capacity.

Of the general population, the elderly are especially vulnerable to respiratory problems, and they can more easily develop respiratory failure, secretion retention and respiratory infections [1,6,7]. This happens due to several factors, including changes in the structure of the upper airways. With decreased pharyngeal muscle support, the risk of aspiration of food content increases due to the decrease in the elastic properties of the lung tissues, the decreased tidal volume of the chest wall and increased respiratory rate [6,7]. Elderly patients with COVID-19 are at greater risk of progressing to a critical disease status [2]. One study reported that between 4% and 11% of all patients with COVID-19 (>30,000 patients) required hospital impatient, with the most severely ill patients transferred to the intensive care unit (ICU) [8]. Despite all these age-related changes making the elderly more vulnerable to the onset of respiratory diseases, whether acute or chronic, the main challenge is to assess the signs and symptoms earlier, making the prognosis more favorable [9].

We can then consider obstructive respiratory diseases and restrictive diseases. The latter is characterized by a reduction in the total lung capacity in the performance of gas exchanges due to the decrease in the expansion of the lung parenchyma [7]. In this type of pathology, there is a progressive deterioration of the ventilatory function, which can lead to chronic respiratory failure and associated hypoventilation. The accumulation of secretions in the airways increases the risk of deterioration due to the inflammatory process associated with its presence, increasing the risk of infection [10]. To prevent airway obstruction, its deterioration and infections, and to optimize lung function, it is imperative to promote/maintain airway permeability [10] using airway cleaning techniques with varied volumes, pulmonary pressures, and expiratory flow [11,12,13].

Effectively, respiratory rehabilitation is fundamental in an increasingly aging society, in which, due to demographic, socioeconomic and environmental factors, the prevalence of respiratory diseases is high [14]. In respiratory rehabilitation in this type of patient, and particularly in those with COVID-19 infection, it is essential to implement self-management strategies. Studies have shown that self-management interventions improve patients’ health status and decrease their risk for hospitalization and emergency department visits [15]. Self-management interventions are structured, multidisciplinary and personalized to achieve clinical goals and to engage and enable adoption of positive health behaviors and development of self-management skills and mastery of their disease [15].

The general objective of this study was to train respiratory patients, so that at home they are able to promote hygiene and airway permeability, improve their ventilatory capacity and prevent complications. As specific objectives: to educate and train the user about the fundamentals and purposes of the techniques, exercises, and strategies of the intervention program until reaching a satisfactory level of knowledge and execution of the exercises; to check compliance with the program; and to evaluate its impact on the users’ lives, 15 days after admission.

## 2. Materials and Methods

### 2.1. The Study Design

An observational, analytical, and prospective pilot study was proposed.

A scientific methodology was followed for this quantitative study by identifying the needs inherent to the studied population concerning self-care and training of participants in the face of acute and chronic conditions, as well as preventing their repetition.

For the implementation of the program, an intervention plan was developed, focused on chronic and acute respiratory patients. This plan was directed towards the patient and did not involve caregivers.

Teachings were part of the functional respiratory re-education programs implemented in the context of hospitalization, teaching them about their pathology, and the interventions that the participants could replicate autonomously at home, providing them with the means to manage their chronic illness or enhance their recovery in case of acute illness. This pilot study considered patients admitted to a hospital with a diagnosis of respiratory disease, acute or with an episode of acute respiratory illness due to chronic illness, which was conscious and oriented in person, time, and space, constituting a sample of convenience.

The dependent variables used in the study for the respective analysis after implementation of the program were: knowledge acquired after teaching during hospitalization; Barthel index at clinical discharge and 15 days after discharge; and degree of dyspnea during hospitalization and 15 days after discharge.

### 2.2. Procedure

This intervention program aims to create a set of exercises and techniques that can be replicated autonomously by patients, allowing them to maintain good hygiene and airway permeability and improve their capacity to execute their daily routines. This was implemented during the respiratory rehabilitation sessions given to admitted patients of the respiratory area, focusing on the knowledge they must acquire, as well as the techniques they can and must master. At the end of hospitalization, the knowledge and skills acquired were evaluated, and 15 days after discharge, a telephone interview was conducted to assess the short-term impact this program had on patients in terms of functionality and independence.

During all the functional respiratory re-education sessions, an approach to pathologies, their physiological characteristics, consequences, signs, and symptoms of aggravation were discussed with patients, trying to make the participants understand the importance and relevance of the techniques/exercises taught to transform them in an asset of the rehabilitation process.

The program consisted of a first phase (during hospitalization) with two exercise sessions (Table 1). Subsequently, the second phase began upon hospital discharge (Table 2). Finally, the third phase began at the patient’s home fifteen days after hospital discharge (Table 2).

At the end of this session, the checklist (Appendix A) was applied for the first time and the dyspnea scale was used to measure the results of the initial intervention. The teachings throughout this program were an integral part of it, maintaining the interventions previously described in the following sessions until the moment of discharge. A checklist was prepared to assess the degree of knowledge and proficiency acquired by the participants during the application of the program. These were based on the selected teachings and intended to evaluate them in two aspects: the knowledge acquired by the participants, whether they were adequate or not, and the technical competence of the participants in replicating them. The evaluation of these aspects was subjective. It was up to the researcher to perform timely questionnaires during functional respiratory re-education sessions and through direct observation of the participants during the execution of the exercises. These were applied at two points: at the end of the second session and at discharge.

In addition to the checklist, a telephone interview was also prepared to be applied fifteen days after discharge, which served as a final assessment tool. The questions were designed to understand participants’ compliance with the program, whether they considered that in-hospital education had been sufficient and if it could be improved, and finally, the impact that the program had on functional capacity, episodes of dyspnea, and elimination of bronchial secretions. The telephone interview was conducted 15 days after discharge, with six questions:Were you able to comply with the recommended exercise program? If not, what are the reasons?Were the teachings you received at the hospital enough?How could the teachings be improved?Do you feel improvements reflected in the decrease of bronchial secretions?Have you had frequent episodes of dyspnea?Has your ability to perform daily living activities improved?

During the implementation of the program, the scale chosen to assess dyspnea was that used by the Medical Research Council, modified during hospitalization (in the first session, at the end of the second and before discharge) and integrated into the final evaluation interview.

The instrument chosen to assess the participants’ functional capacity was the Barthel Index [16]. This is a scale that translates the result of the evaluation performed by the professional on the patient’s capacity for self-care, already validated in the country in which this study was conducted. This was applied at three points: in the first session, before discharge and fifteen days after discharge, integrated in the telephone interview.

### 2.3. Ethical Considerations

Procedural measures, such as the confidentiality and protection of the participants’ data involved in the intervention program, have been complied with. The project of the intervention program was approved by all the mediating entities: the University of Évora, the administration council of the hospital unit and the person in charge of that unit’s service. In addition, a request was submitted to request an evaluation of the project from the Ethics Committee—Health and Welfare Area of the University of Évora, which obtained a positive opinion. All patients included in the sample were given informed consent, the right to confidentiality, anonymity, and self-determination. In this way, it was guaranteed through a written commitment that the data collected would always be anonymous and confidential, having been encrypted and used only for this purpose.

## 3. Results

In this pilot study, the defined target population was patients admitted to a hospital between 19 September and 22 November 2017, with a diagnosis of acute respiratory disease or an acute episode of chronic respiratory disease, who were conscious and oriented in person, time, and space. Five participants were included to whom the intervention program was applied: three males and two females, aged between 39 and 76 years old. Of these, four were diagnosed with CAP and one with acute COPD. Regarding underlying diseases, one patient reported recurrent respiratory infections, five patients reported arterial hypertension, and two reported type II diabetes and moderate alcohol habits. However, the major factor that aggravated the clinical situation was aging. As for the length of stay, it was found that four participants had five days of stay and only one had four days.

The older participants in this pilot study had more difficulties, taking longer to assimilate the concepts, demonstrating that they had acquired the desired knowledge only in the last sessions. In contrast, the younger participants quickly incorporated the concepts within the first two sessions. However, despite this, all five participants achieved the intended objective. That is, they all demonstrated basic knowledge about the means to stabilize their disease, symptoms, and signs of worsening.

The participants, according to their pathology, were trained with nine predetermined teachings, techniques, and exercises which they would be able to perform independently. To check their understanding and proficiency in the teachings conducted during the second session of functional respiratory re-education and the last session before discharge, a checklist (Appendix A) created for this purpose was applied.

In the first application of the checklist, after teaching the techniques of rest/relaxation and awareness/breath control, it was found that all patients subject to the program reached a satisfactory level of understanding of their objectives and procedures and that this remained in the last assessment before discharge. Indeed, dyspnea is the principal limiting agent of the independence and functionality of respiratory patients. Knowledge of all the techniques and knowledge associated with the teaching of energy conservation measures is of great relevance, both in rehabilitating these patients and in the prevention of crises and accidents. These energy-saving measures were easily learned by all users in both evaluations, except for the MR patient who initially showed difficulties; at the end of the hospitalization at the last application of the checklist, however, he demonstrated all the necessary knowledge.

The knowledge acquired after the teachings conducted during hospitalization, and the purposes intended in the intervention program, were: to reduce psychological and muscular tension; to prevent and correct ventilatory defects; to promote airway hygiene and permeability; and to empower the user to perform daily living activities (Figure 1).

In the first assessment, none of the participants applied the strategies and techniques for energy conservation regularly and proficiently. Only during hospitalization and with the reinforcement of training, in respiratory functional re-education sessions, and opportunistically (for example, in the performance of hygiene), did the participants begin to apply them correctly and regularly, as verified in the last evaluation before discharge.

The degree of dyspnea is a fundamental factor for the independence and functionality of the patients, so it was evaluated at four points: at admission, at the end of the second session of functional respiratory re-education, at discharge, and 15 days after discharge (Table 3).

All participants started with high degrees of dyspnea, which decreased during hospitalization. Although all of them improved significantly, none reached grade 0 until discharge. On the other hand, only one patient had grade 2, and all the others returned home with grade 1. With the application of this program, at 15 days after discharge, three patients decreased their degree of dyspnea.

To assess the degree of dependency/functionality according to the evolution of the treatment/intervention of the rehabilitation nurse specialist throughout the pilot study, the Barthel Index was used (Figure 2). Effectively, upon admission, two participants were classified as dependent, and the rest were classified as partially dependent, with scores between 55 and 60. At discharge, all were classified as independent with scores between 90 and 100, except the MR participant, who remained partially dependent, with a score of 85. Thus, we can say that there was a significant recovery during hospitalization that was not unrelated to the program implemented, motivating and enabling participants to contribute to their recovery outside the functional respiratory re-education sessions. Equally or more important was the fact that, at the last evaluation 15 days after discharge, none of these participants had a lower score than the previous evaluation at the hospital, maintaining the score.

## 4. Discussion

In this pilot study, all participants started with high degrees of dyspnea, which decreased during hospitalization and after discharge. The scientific literature advises the use of systematic assessment instruments (physical examination and complementary means of diagnosis) of the patients before and after the implementation of the intervention plan, considering the functional capacity, quality of life (Qol), respiratory and muscular function [17,18]. In these cases, the most common symptomatology includes dyspnea, fever, cough, myalgia, and fatigue. The respiratory symptoms related to COVID are very similar, which makes it essential to manage isolated patients properly to offer the best possible treatment for them and as soon as possible [19]. The set of strategies and techniques for energy conservation are intended to facilitate the performance of life activities with less fatigue or shortness of breath in respiratory patients, making them more autonomous [20].

In this context, the role of prevention, treatment and rehabilitation of the rehabilitation nurse specialist stands out, assuming a privileged place within the multidisciplinary team, either because the set of specialized knowledge that it gathers, its proximity, trust, or a holistic approach [1,5,21]. The rehabilitation nurse specialist should provide education on paced breathing and relaxation techniques to optimize respiratory function [20].

After our program, at discharge, all patients were classified as independent with scores between 90 and 100 (Barthel Index), except the MR participant, who remained partially dependent, with a score of 85. Respiratory rehabilitation is one of the most studied areas, due to the high dependence that respiratory diseases can cause, with negative repercussions on autonomy and Qol [1]. Like these patients, COVID-19 patients show significant physical limitations, with absolute improvements during the rehabilitation program [3]. They may fatigue easily and require frequent rest breaks [20]. Oxygen saturation levels should be maintained above 95%, and patients should be closely monitored for shortness of breath to assess activity tolerance [20].

In our pilot study, with five participants aged between 39 and 76 years, four had a diagnosis of CAP and one of COPD. The profile of the respiratory patient has changed over time, influenced by economic, social, environmental, and demographic factors. Infectious respiratory diseases are transmitted from person to person by air, so their frequency is directly proportional to higher population concentration and lower health and hygiene conditions [9]. COPD is one of the most relevant obstructive respiratory pathologies regarding the intervention of the health professional [22], having been one of the main diagnoses of the target population to whom the present intervention program was applied. The development of these programs of rehabilitation nursing, promoting teaching, demonstration, and training of techniques, promoting self-care and continuity of care in different contexts, makes knowledge of COPD and the optimization of emotional capacity possible through the dominion over the disease, with moderately large and clinically significant improvements [23].

In more developed countries, the incidence and hospitalization for CAP is high, and these values increase among children, the elderly, and during the winter [6,9]. The treatment of CAP is not only dependent on antibiotics but also on the appropriate use of oxygen, adequate nutritional support, prevention, or treatment of metabolic disorders, gradually improving the Qol of these users [1,24]. Indeed, the rehabilitation programs that aim to improve the function of the respiratory system should be included in the management of these patients, such as those with COVID-19 [18]. Patients can be included in a similar process to our study. After starting a respiratory rehabilitation program during hospitalization, they are sent to their homes and monitored through a hybrid telerehabilitation model. Hybrid models that include assessment, exercise, a combination of virtual exercise training, education, and self-management can optimize exercise safety and training effectiveness while decreasing the risk of disease transmission and infection rates [15].

Respiratory rehabilitation is one of the most studied areas due to the high dependence that respiratory diseases can cause, with negative repercussions on autonomy, but also due to epidemiological data, in an industrialized and aging society, highlighting the need for the rehabilitation nurse specialist intervention in the recovery/rehabilitation of these patients [1]. At the clinician level, rehabilitation nurse specialists must support family caregivers during the COVID-19 pandemic, with intervention models based on patient and family education during hospitalization and after clinical discharge [20]. Interventions such as incentive spirometry should be emphasized, and patients should be instructed on strategies such as diaphragmatic and pursed lip breathing [20].

Early intervention [2], regular follow-up and rehabilitation guidance at discharge for patients with respiratory diseases can improve vital capacity and cardiopulmonary endurance.

## 5. Conclusions

Thus, we can conclude that this intervention program allowed us to develop a proposal of action based on the training of the respiratory patient. In the long term, these participants need to be accompanied by a rehabilitation nurse specialist at home to reinforce the teachings and training provided and to assist these participants in the management of their disease. At the hospital level, it is essential to involve the entire nursing team in respiratory rehabilitation programs to achieve continuity of care, since the average hospital stay is reduced.

In effect, this program positively impacted on the participants’ rehabilitation process, allowing them to be active agents in their own rehabilitation, enabling them with the means to prevent complications. The lack of support that these participants felt after discharge is accurate, and it is necessary to develop support and follow-up mechanisms with the work of the rehabilitation nurse specialist in the community, contributing to the reduction of the number of emergency episodes, the number of hospitalizations and the number of deaths from respiratory disease.

The intervention program was designed and applied to patients with respiratory diseases. The COVID-19 symptomatology and comorbidity are similar, so it would also impact these patients. We emphasize the importance of respiratory rehabilitation after infection by COVID-19, based on their needs when hospitalized and after discharge, given the improvement in the degree of dyspnea and functionality of these patients.

## 6. Limitations of Study

Some limitations have been included in this study. The use of a pilot study is a limitation because it does not allow the generalization of data. More studies are needed with a larger number of patients and longer duration with follow-up observation. Another current limitation is the scarcity of qualified health professionals trained in home rehabilitation programs, so it was only possible to conduct a telephone interview 15 days after clinical discharge. However, the six questions asked made it possible to assess the level of knowledge acquired by the patients.

## Figures and Tables

**Figure 1 ijerph-19-14365-f001:**
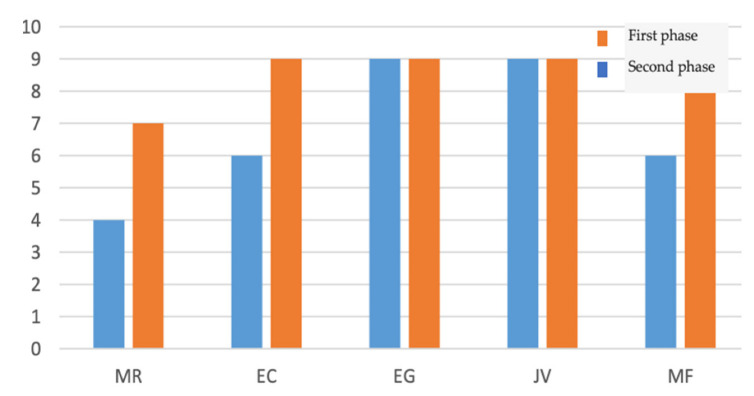
Results of the knowledge acquired after the teachings conducted during hospitalization (six answers).

**Figure 2 ijerph-19-14365-f002:**
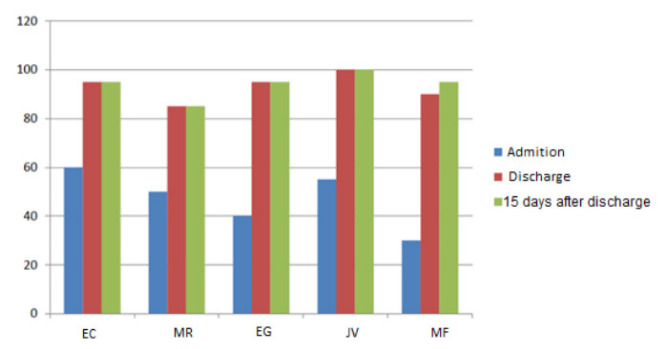
Barthel index score of the 5 participants in the sample, at admission, at discharge and 15 days after discharge.

**Table 1 ijerph-19-14365-t001:** First phase.

Session	Objectives	Teachings/Training	Evaluation
First	Reduce psychic and muscle tension, reducing muscle overload	Rest and relaxation techniques (twice a day) Breath awareness and control (twice a day) Diaphragmatic breathing (twice a day) Incentive spirometry (twice a day) Global costal opening (twice a day) Diaphragmatic reeducation with resistance (twice a day) Teach directed cough (twice a day) Huffing (2 times per shift)	Application of the Barthel Index
	Ensure airway permeability		
Second	Re-educate the efforts	Energy conservation techniques (twice a day and opportunistically) Training of activities of daily living (opportunistically during the day)	Application of the dyspnea scale (mMRC)

**Table 2 ijerph-19-14365-t002:** Second phase and third phase (15 days after discharge).

Second Phase	Third Phase
Second application of the check list, to measure knowledge, assimilated techniques, and evolution, during hospitalization	Telephone interview 15 days after discharge
Third application of the dyspnea scale (mMRC)	Application of the dyspnea scale (mMRC)
Second application of Barthel Index	Application of the Barthel Index

**Table 3 ijerph-19-14365-t003:** Degree of dyspnea, during hospitalization and 15 days after discharge.

Participants	1° Day Impatient	Half Stay	Discharge	15 Days after Discharge
EC	3	2	1	1
MR	4	3	2	1
EG	3	2	1	0
JV	3	2	1	0
MF	4	3	1	1

## Data Availability

No new data were created or analyzed in this pilot study. Data sharing is not applicable to this article.

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
