# Peer review of "A Pilot Study on a Nurse Rehabilitation Program: Could It Be Applied to COVID-19 Patients?"

_ijerph, 2022, doi:10.3390/ijerph192114365_

Round 1

Reviewer 1 Report

Dear Editors,

Thank you have the chance to review the study; the article has some issues below,

1. Introduction section and conclusions section, so many sentences high similar with published papers and web sites studies.

2. Materials and Methods section, the Individuals under Study section, and the Procedure which teaches the patients the detailed process is unclear; please address more. In addition, the intervention program related to respiratory rehabilitation sessions should add references or stand protocol support. Furthermore, should authors address more how to ensure the patients did get the correct intervention program only by telephone tracking.

3. On Page 5, the study stated that “At the end of this session, the checklist (Appendix A)…” yet, I still could not find the Appendix from the article.

4. For more contribution, telephone interviews should add related references or expert reliability and validity. In addition, the telephone interview context and guidelines lack related QOL, thus how to ensure that the results point to the aims.

5. On Page 6, the study lack a qualitative or quantitative analysis method.

6. How to recruit COVID-19 patients and screen suitable patients, such as excluding or including them in the program; please address more.

7. What are the MR, EC, Eg, JV, MF five patients related underlying diseases? Please clarify.

8. Due to unclear study process, and QOL measurement tools, thus the conclusions section may need to reword.

Thank you.

Author Response

Thank you for your comments. Attached is the file with the answers.

Reviewer 2 Report

First of all, congratulations for the work done, it is a research of great current interest. After reviewing the manuscript, the following points could be improved:

In the introduction section, the justification of the study should be improved with regard to the need for self-care of the patient and specifically the covid respiratory patient.

It is essential for the publication of this article to have a methodology section explaining the method used to arrive at the results offered, the variables, statistical tests, significance and tables of results.

Finally, I think it would be interesting to include a section on the "limitations of the study" in the manuscript. 

Best regards 

Author Response

(The authors gave the same response as above.)

Reviewer 3 Report

The authors describe a pilot study regarding the implementation of respiratory rehabilitation programme to five patients admitted to a hospital. Pulmonary rehabilitation is important in healthcare, and in particular considering the prevalence of respiratory diseases and increasing levels of pollution affecting all age groups. Explorations are also necessary in view of the ageing population and the associated risks of respiratory diseases for the older adults. The issue being addressed is of value. However, in the current manuscript, there are major issues that the authors need to address and clarify.

1.

More details are expected in the materials and methods section (p.3), for example:

ln 93, there is only one brief statement for section 2.1 (The study design). Details regarding the research question and how the design of the study was planned to achieve the objectives should also be described with rationales.

Ln 95-97 (Section 2.2 Individuals under study), what is being conveyed by the statement “a scientific methodology was followed by identifying the needs inherent to the studied population…” is unclear. What was the methodology referring to? Although you have mentioned in the Results Section (ln 176-182) for some participant details, the inclusion criteria, as well as the development, implementation and intervention details should be more covered in the materials and methods section.

Ln 160-162, regarding the Barthel Index, would authors consider provide any related study reference on the use to assess functional capacity or a formal reference on the assessment instrument.

Table 1 about the teaching content (p.4), some items are indicated with the frequency in each day, but some are not. Are there any duration details for the items or the sessions?

2.

Results and discussions should be presented in more detail, for example, what are the qualitative findings regarding the questions in the telephone interview as mentioned in ln 149-156? The limitations of the study should also be discussed more as a separate section, instead of the statement in ln 316. These also include the assessment methods with the appendix checklist and telephone interview, or any enhancement or development plan from the current study?

3.

The introduction should be strengthened with a stronger background. In addition to the significance of respiratory rehabilitation, there should be more discussion and references of related studies regarding the use of intervention programmes in rehabilitation and assessment of Qol. Similarly as commented above, discussions on the use of assessment tool and Barthel Index in the literature would be expected for the strengthening of the introduction section.

4.

The writing needs to be edited and improved. There are grammatic and spelling errors observed which obscure the meaning. 

Author Response

(The authors gave the same response as above.)

Round 2

Reviewer 1 Report

Dear Authors,

Thank you for your response. Please add my initial comments and revise your manuscript.

Reviewer 2 Report

Congratulations on the revision, it has substantially improved the manuscript. The research done is of great interest to the reader.

Regards

Reviewer 3 Report

The authors incorporated my comments. The added information and limitation section are acknowledged. I have an observation for the author team to consider. In ln 307-308, the following statement is cited with reference item 18 (ln 416-418), which is the study regarding the effectiveness of nonpharmacologic airway clearance therapies in hospitalized patients, by Strickland et al. I think the correct reference for the statement in ln 307-308 should be based on reference item 19 (ln 419-421)? Please check should it be referenced from the study by Gonzalez-Gerez et al., which is about short-term effects of a respiratory telerehabilitation program in confined COVID-19 patients.

Ln 307-308:

“Indeed, the rehabilitation programs that aim to improve the function of the respiratory system should be included in the management of this patients such as those with COVID-19 [18].”

Ln 419-421, Reference 19:

Gonzalez-Gerez, J.J.; Saavedra-Hernandez, M.; Anarte-Lazo, E.; Bernal-Utrera, C.; Perez-Ale, M.; Rodriguez-Blanco, C. Short-Term Effects of a Respiratory Telerehabilitation Program in Confined COVID-19 Patients in the Acute Phase: A Pilot Study. Int J Environ Res Public Health. 2021 18, 7511.